# High Genetic Diversity of an Invasive Alien Species: Comparison between Fur-Farmed and Feral American Mink (*Neovison vison*) in China

**DOI:** 10.3390/ani11020472

**Published:** 2021-02-10

**Authors:** Lina Zhang, Yan Hua, Shichao Wei

**Affiliations:** 1Eco-Engineering Department, Guangdong Eco-Engineering Polytechnic, Guangzhou 510520, China; zlntiantang@163.com; 2Guangdong Provincial Key Laboratory of Silviculture, Protection and Utilization, Guangdong Academy of Forestry, Guangzhou 510520, China; gaf@sinogaf.cn

**Keywords:** *Neovison vison*, microsatellite, biological invasion, mink farming, genetic variation

## Abstract

**Simple Summary:**

The American mink (*Neovison vison*) is one of the best-known and most widespread invasive species in China and worldwide. To investigate the genetic characteristics and increase comprehension of the invasiveness process for this species, we compared the genetic characteristics of farmed and feral populations in northeastern China using mitochondrial DNA sequences and microsatellite loci. We found a relatively high diversity among the feral populations that was as high as that of the farmed mink. This demonstrated that high genetic diversity promotes the invasiveness and rapid evolution in the wild.

**Abstract:**

Genetic characteristics play an important role in alien species for achieving high adaptation and rapid evolution in a new environment. The American mink (*Neovison vison*) is one of the best-known and most widespread invasive species that has successfully invaded the Eurasian mainland over quite a short period, including most parts of northeastern China. However, genetic information on farmed and feral American mink populations introduced in China is completely lacking. In this study, we combined mitochondrial DNA sequences and polymorphic microsatellites to examine the genetic divergence and genetic diversity of farmed and feral American mink populations. Our results suggest that there is admixture of individuals of different genetic characteristics between farmed and feral populations of mink. Furthermore, the genetic diversity of both farmed and feral American mink populations was high, and no bottleneck or population expansion was detected in most of the populations. These findings not only highlight the genetic characteristics of American mink in northeastern China but also contribute to the general understanding of the invasiveness of farmed species.

## 1. Introduction

Genetic characteristics play an important role in alien species occupying new environments and expanding their distributions [1,2,3]. Usually, introduced populations are founded by a limited number of individuals, and natural selection and genetic drift will result in lower genetic variation than in native populations [4,5,6]. However, observations of some alien species invasions have shown evidence of introduced populations without reduced genetic variability [3,5,7,8]. It is possible that genetic diversity in introduced populations is not lost when large numbers of animals are present or multiple introductions occur (propagule pressure) [9,10].

The American mink (*Neovison vison*), a semi-aquatic species of mustelid endemic to North America, was brought to Europe, Asia, and South America for fur farming operations. In Europe, the American mink was first introduced as a furbearer via Russia in the 1920s [11], followed by England, France, Germany, Iceland, Ireland, Norway, Poland, Scotland, Sweden, and other parts of Europe. Invasive American mink populations have higher levels of genetic diversity from genetically diverse sources in Poland [12] and Spain [13]. In Asia, American mink was initially introduced to the eastern and southern former Soviet Union, including Sverdlovsk, Irkutsk Oblasts, Yakut, Magadan, and areas adjacent to China, such as Amur Oblasts, Khabarovsk, and Primorsky Krai [14]. The escapees or deliberately released individuals founded discontinuous populations due to their excellent ability to colonize new habitats [15]. In eastern Asia, it is possible to rapidly establish growing feral populations of this species in aquatic ecosystems [16].

In China, since the middle of the last century, a large-scale fur farming industry of American mink has thrived in the north, including in Heilongjiang, Jilin, Liaoning, Shandong, and Hebei provinces [17]. In 1978, the number of breeding American mink reached 750,000 females. In 2014, more than 80 million mink were bred on farms in China, ranking first in the amount of both breeding farms and fur production worldwide [18]. Especially for the region of Mohe County and adjacent counties, there were a large number of escapees that successfully invaded large parts of northeast China, and feral mink from Russia also migrated into the area [16]. However, genetic information on farmed and feral American mink populations introduced in China is completely lacking. Thus, there is still confusion as to whether there is a similar high genetic diversity level between farmed and feral populations, whether the process of escaping is ongoing, and how mink farms have affected feral populations in the present and past years.

In this study, we investigated the genetic variability in both farmed and feral American mink in northeastern China by mitochondrial DNA sequences and by characterizing polymorphic nuclear microsatellites. We hypothesized that domestic mink are escaping from farms and becoming feral and that the process is ongoing. Thus, we hypothesized that there is a relatively high diversity among the feral populations and that the samples that come from farmed or feral populations would cluster together.

## 2. Materials and Methods

### 2.1. Sampling

From 2011 to 2012, a total of 30 muscle tissue samples from 2 sites were collected from captive American mink on fur farms, and another 32 feral mink from 4 sites were captured in the areas of Mohe County and Tahe County along the branches of the Heilongjiang River in China (Figure 1). Although the number of samples is low, 15 to 20 individuals per genetic cluster are sufficient to accurately estimate genetic diversity [19]. Additionally, there are different colour types of mink, but these did not affect the cytochrome b (*Cyt-b*) gene or the microsatellites used in this study [20]. Tissue samples from captive mink were collected during the skinning process, and tissue samples from feral mink were obtained from newly hunted animals. All tissues were stored in 95% ethanol before transportation to the laboratory and stored at −20 °C until DNA extraction.

### 2.2. Laboratory Analyses

Genomic DNA was extracted from tissue samples using a DNeasy kit (Qiagen) following the manufacturer′s protocol. We used 12 polymorphic microsatellites selected from Vincent et al. [21] and amplified them under the following conditions: denaturation at 94 °C for 5 min, 30 cycles of denaturation at 98 °C (10 s), annealing at 50–68.5 °C (30 s), and extension at 68 °C (20 s), followed by a final extension at 68 °C (20 min). PCR amplification was carried out in 20-µL reactions containing 1 × PCR buffer containing 50 mM Tris-HCl (pH 8.0), 25 mM KCl, 0.1 mM EDTA, 1 mM dTT, 0.4 mM of each dNTP (TOYOBO), 0.2 µM of each forward primer (labelled with Hex, Fam or Ned fluorescent dyes) and reverse primer, 0.4 U units of KOD FX Neo DNA polymerase (TOYOBO) and approximately 50 ng of genomic DNA. The PCR products were separated using an ABI 3700 Prism automated sequencer and scored using GeneScan 3.7 and Genotyper 2.5 (Applied Biosystems).

An 890 base pair fragment of mitochondrial DNA containing the cytochrome b gene was amplified for each sample with primers L12616/H01177 [22]. PCR was performed in 20-µL reactions containing 1 × PCR buffer (50 mM Tris-HCl (pH 8.0), 25 mM KCl, 0.1 mM EDTA, 1 mM dTT), 0.4 mM of each dNTP (TOYOBO), 0.2 µM of each forward and reverse primer, 0.4 U units of KOD FX Neo DNA polymerase (TOYOBO), and approximately 50 ng of genomic DNA. PCR was run with an initial denaturation at 94 °C for 10 min, 30 cycles of denaturation at 98 °C (10 s), annealing at 55 °C (30 s), and extension at 68 °C (1 min), followed by a final extension at 68 °C (30 min). PCR fragments were sequenced in both directions using an ABI 3700 Prism automated sequencer.

### 2.3. Genetic Diversity

We used MICRO-CHECKER to examine the populations for the possible presence of null alleles and allelic dropout [23]. Note, “population” refers to sampling sites throughout the following text. We analysed the number of alleles (*N_A_*), expected heterozygosity (*H_E_*), observed heterozygosity (*H_O_*), and deviations from Hardy–Weinberg equilibrium (*HWE*) and linkage disequilibrium (*LD*) for each locus or population using ARLEQUIN 3.5 [24]. We used Bonferroni corrections for multiple comparisons to find critical significance levels for both tests (equivalent to *p* < 0.05, [25]). Moreover, other genetic diversity parameters, including allelic richness (*AR*), inbreeding coefficients (*F*_IS_), and polymorphic information content (*PIC*), were calculated using FSTAT [26].

We sequenced and aligned the *Cyt-b* gene sequences of all samples using MEGA 6.0 [27]. We identified unique haplotypes and calculated the number of haplotypes (*h*), haplotype diversity (*Hd*), and nucleotide diversity (*π*) for farmed and feral populations using DnaSP 5.0 [28].

### 2.4. Individual Assignments

We used the microsatellite data set to determine the individual assignments of all samples using STRUCTURE 2.3.4 [29]. We used the admixture model with correlated allele frequencies and 1,000,000 Markov chain Monte Carlo (MCMC) iterations with 100,000 burn-ins. We performed 10 independent simulations and checked for the consistency of the runs. We determined the most likely *K* by following the simulation method of Evanno et al. [30], and the optimal number of clusters was estimated to be between *K* = 1 and *K* = 6 using the web-based software STRUCTURE HARVESTER v0.6.8 [31]. Additionally, the genetic structure based on the *Cyt-b* gene and microsatellites may provide a different view due to the higher mutation rate of microsatellites compared to mtDNA, incomplete lineage sorting, recent admixture, and strong male-biased dispersal in the American mink [32]. Thus, we also constructed Bayesian inference of phylogenetic trees with BEAST v2.4.4 [33] using 50,000,000 MCMC iterations, with sampling per 5000 iterations. Three independent analyses were checked for the convergence of the MCMC and effective sample sizes (above 200) in TRACER v.1.7 [34]. We used the program DENSITREE v.2.2.6 [33] to visualize the trees after discarding the first 10% of each MCMC chain as burn-in. The evolutionary relationships among farmed and feral mink haplotypes were also inferred from the maximum parsimony-based median-joining network that was calculated and drawn using NETWORK 4.5.1.6 (http://www.fluxus-engineering.com) (accessed on 30 December 2020).

### 2.5. Population Demography

The samples from a single location were defined as a population. We analysed the demographics of the six populations using three methods. First, a genetic bottleneck was investigated using the program BOTTLENECK 1.2 [35]. Significance was assessed using the Sign and the Wilcoxon sign-rank test to test the allele frequency distributions for shifts from the equilibrium L-shape [36]. Second, neutral tests, including Tajima′s *D* and Fu′s *Fs*, were conducted for farmed and feral populations by ARLEQUIN with significantly negative values suggesting expansion. Furthermore, the mismatch distribution was analysed by examining the smoothness of the observed distributions and the degree of compatibility within the observed and simulated patterns.

## 3. Results

### 3.1. Genetic Diversity

We did not detect the presence of null alleles or allelic dropout in the microsatellite data. All analysed loci were highly variable with 3 to 15 alleles per locus with a mean estimate of 9 overall loci (Table 1). For the whole sample set, the ranges of the *AR*, *H_E_*, and *Ho* values were 4.08–4.94, 0.61–0.71, and 0.50–0.63, respectively (Table 2). The mean number of alleles and the allelic richness of the farmed mink (6.33–6.58 and 4.64–4.94, respectively) was slightly higher than that of the feral populations (4.08–5.00 and 4.08–4.50, respectively) (Table 2). The heterozygosity indices for the farmed mink (*H_E_* = 0.66–0.71, *H_O_* = 0.59–0.60) were slightly higher than for the feral populations (*H_E_* = 0.61–0.66, *H_O_* = 0.50–0.63) (Table 2). All populations showed positive *F*_IS_ values (0.06–0.21) (Table 2).

Sequences of the *Cyt-b* gene from all samples (*n* = 62) yielded 23 haplotypes containing 85 polymorphic sites (8 singleton variable sites and 77 parsimony informative sites). We detected 11 haplotypes in farmed mink and 15 haplotypes in feral mink (Table 2). In particular, NV2 was the most common haplotype, which was observed in 14 farmed and 10 feral samples. In contrast to the microsatellite data results, the *Cyt-b* gene diversity was higher for the feral populations (*Hd* = 0.71–1.00, *π* = 0.01–0.02) than for the farmed mink (*Hd* = 0.24–0.82, *π* = 0.01; Table 2).

### 3.2. Individual Assignments

Our clustering results supported the grouping of American mink samples from northeastern China into two genetic clusters. Generally, the first genetic cluster corresponded to samples of farmed animals, and the other cluster corresponded to feral samples. Some samples showed significant signs of admixture, as two farmed individual and eleven feral samples were assigned to their cluster with likelihoods less than 0.7 (Figure 2).

The Bayesian phylogeny displayed five branches in the American mink from north-eastern China (Figure 3). The population of “Feral” was separated from other populations. The populations of “Farmed 6” and “Feral 3” were mixed by 1 and 3 individuals, respectively. The population of “Feral 2” clustered with the population of “Farmed 5”. Two individuals of “Farmed 5” were assigned to the genetic cluster formed by the “Feral 1” population (Figure 3). The median-joining network produced a distribution of haplotypes, and it should be noted that the 6 feral haplotypes clearly differed from the remaining haplotype group (66 substitutions) (Figure 4).

### 3.3. Population Demography

Three methods were used to infer the demographic histories of the six populations. For all populations, Tajima′s *D* was not significantly negative (Table 2). For the two farmed mink, Fu′s *Fs* was significantly negative (Table 2). For the four feral populations, Fu′s *Fs* was positive for population “Feral 2”, significantly negative for population “Feral 1”, and negative but not significantly for populations “Feral 3” and “Feral 4”. In the mismatch distribution analysis, we did not detect expansion signals in any populations (Table 2). In the genetic bottleneck analysis, the populations “Farmed 5”, “Farmed 6”, “Feral 1”, and “Feral 3” were non-bottlenecked. However, we detected significant bottleneck events in populations “Feral 2” and “Feral 4” (Table 2).

## 4. Discussion

We discovered a similar high genetic diversity between the farmed and feral populations through microsatellite and *Cyt-b* gene analysis. Furthermore, we found that the farmed and feral mink populations in northeastern China could be divided into two genetic clusters. In addition, we did not detect rapid expansion or bottlenecks in most of the populations, which may be due to the large population size or multiple introductions.

Our results show that feral mink in northeastern China exhibits moderate to high genetic diversity. The genetic diversity based on microsatellite data of feral mink in northeastern China (*H_O_ =* 0.50–0.63, *H_E_ =* 0.61–0.66) was comparable with that in other regions, such as Scotland (*H_O_* = 0.55–0.66, *H_E_* = 0.56–0.67) [37], but was slightly lower than that in their native distribution area (*H_O_* = 0.64, *H_E_* = 0.74) [21]. The high genetic diversity may be due to multiple origins, creating the conditions for high adaptable capacity and expanding potential; this aids our understanding of invasion processes and dynamics [38,39].

The genetic diversity of the farmed mink was higher than that of the feral populations according to the microsatellite results; in contrast, there was a much higher mtDNA diversity in the feral mink than in the farmed mink. The inconsistency of the genetic diversity results between the two markers may possibly be because of multiple admixtures at a very early stage in the feral mink, as we found a large number of shared haplotypes in the median-joining network; or it could be due to the different methodological approaches used, which included different information content [40].

Bayesian clustering analysis identified two genetic clusters corresponding to the farmed and feral mink samples. However, the phylogenetic tree showed five branches, four of which comprised samples from different locations. The reason for this could be that the *Cyt-b* gene provides more of a historical view whereas microsatellites provide an understanding of more recent events. Furthermore, the significant signs of admixture and the samples from different locations clustered together in the phylogenetic tree may illustrate that the process of farmed mink escaping into the wild is ongoing. Two feral mink pedigrees of different origins led us to rethink the sources of feral mink in this region. This could be the result of the feral mink originating from farmed mink that had adapted to the environment under selection in the past, or, recently, some escaped farmed mink hybridizing with the feral mink.

The rapid expansion hypothesis was rejected in both the farmed and feral mink populations, and bottlenecks were only detected in populations “Feral 2” and “Feral 4”. This could be explained by the large numbers of animals present or multiple introductions that occurred as well as eradication control measures in China. A similar situation in this species occurred in Spain, where most populations, except for those in Catalonia, were not affected by a bottleneck [13].

## 5. Conclusions

A comparison of the genetic characteristics of farmed and feral populations can contribute to understanding the invasiveness process for different species. Our results suggest that there is a relatively high diversity and an admixture of individuals of different genetic characteristics between farmed and feral populations of mink in northeastern China, which is conducive to increasing the fitness of individuals and potentially contribute to the invasion of American mink. Nevertheless, additional studies should be conducted to more fully understand the genetic variability of the American mink in northeastern Asia.

## Figures and Tables

**Figure 1 animals-11-00472-f001:**
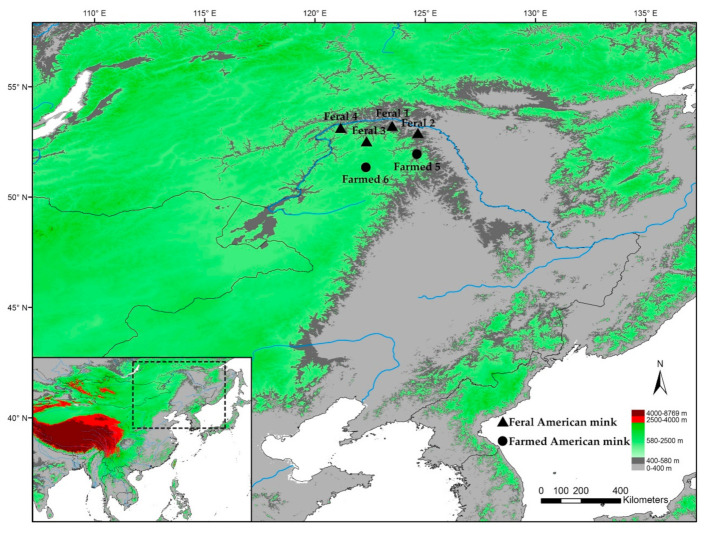
Map of the study area and sampling location of farmed and feral American mink.

**Figure 2 animals-11-00472-f002:**
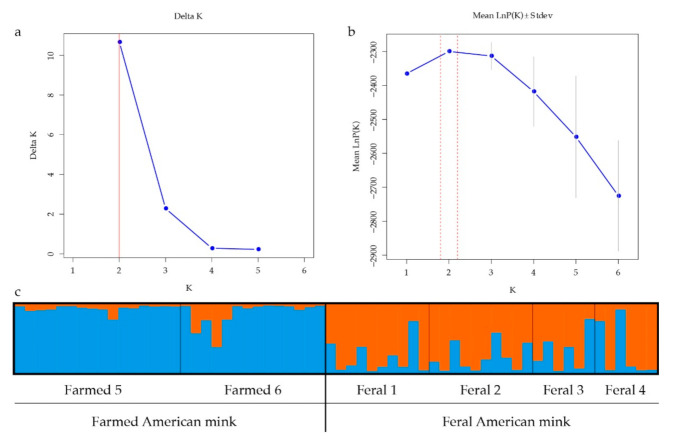
STRUCTURE clustering results deduced from microsatellite alleles within populations of American mink. (**a**,**b**) Δ*K* and Ln *P*(*X*/*K*) values as a function of the *K* values according to 10 run outputs; (**c**) STRUCTURE clustering results at *K* = 2, with different colours representing different clusters, deduced from microsatellite alleles within farmed and feral populations of American mink.

**Figure 3 animals-11-00472-f003:**
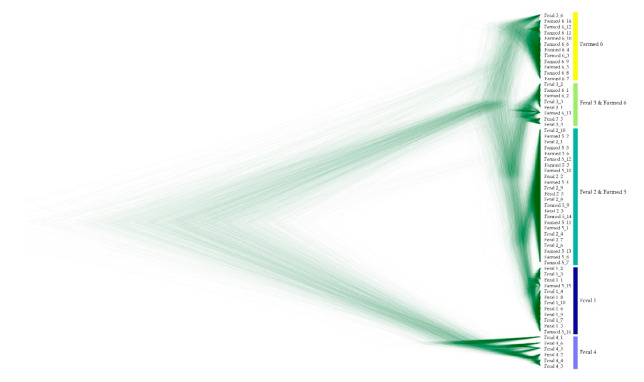
The phylogenetic relationships among *Cyt-b* haplotypes for feral and farmed mink display the whole trees generated from DensiTree.

**Figure 4 animals-11-00472-f004:**
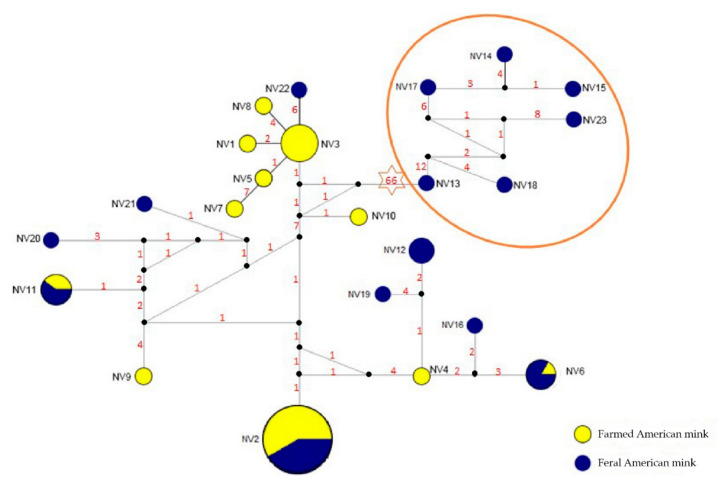
Median-joining network with node sizes proportional to the frequencies of farmed and feral mink haplotypes. The numbers of mutations separating the haplotypes are shown on the branches. The yellow colour represents the farmed mink, the blue colour represents the feral mink, and the small black dot indicates undetected haplotypes. One red area indicates the cluster of 6 feral haplotypes.

**Table 1 animals-11-00472-t001:** Information on the 12 microsatellite loci used in this study. The loci name, primer sequence, repeat motif, annealing temperature (T_A_), numbers of alleles, allele size range, observed (*H_O_*), expected (*H_E_*) heterozygosities, and average polymorphism information content (*PIC*) were included.

Loci	Primer Sequence (5′–3′)	Repeat Motif	T_A_ (°C)	No. Alleles	Size Range	*H_O_*	*H_E_*	*PIC*
Mvi 1271	F: TAA ACA CGG CTC ACT AAC TC R: GTG GTA TGC ACT CAA GGT	(CA)_15_	61.0	6	180–190	0.73	0.73	0.68
Mvi 1272	F: CCT CCC CTT CTC GTG R: TCT TTC TGC TAT TCG GTA AG	(TC)_14_ AT(TC)_4_	60.3	7	165–179	0.50	0.68	0.64
Mvi 1273	F: GCT TAA TTC GTA TAG CAT CCC T R: CCT CCA GAC CTC TAG CAT C	(GGAA)_6_	59.0	15	183–213	0.80	0.88	0.86
Mvi 1302	F: CAT AGG TTC CAG GGA TTA GAA R: ATG CCA TTA CAG TAC GAC TCA	(GT)_17_	64.0	8	204–224	0.44	0.69	0.64
Mvi 1321	F: TTA AAC ACG AGA CCG TAT GTA R: GAA AGT GTG CCA ATT CCT A	(CA)_13_	63.5	15	91–179	0.61	0.85	0.83
Mvi 1322	F: GGC TGA TTA ATA TTT TAC ACA R: CAA AAA CCA CTA CCT CAA	(CA)_12_	50.0	11	160–180	0.62	0.84	0.82
Mvi 1323	F: AAT GGG GGA ATT TAC AGG T R: CTG AAA TAC AAG GGC ATT CTT	(GT)_9_ GC (GT)_4_	60.0	4	104–110	0.29	0.52	0.43
Mvi 1341	F: GTG GGA GAC TGA GAT AGG TCA R: GGC AAC TTG AAT GGA CTA AGA	(CA)_17_	59.0	9	150–166	0.95	0.82	0.79
Mvi 1342	F: TGG GAG TGA GCG GTG AT R: CTG GCC TTC AGT CAG TCT TG	(AC)_14_	68.5	13	131–163	0.47	0.84	0.82
Mvi 1354	F: CCA ACT GGA GCA AGT AAA T R: CAT CTT TGG GAA AGT ATG TTT	(CA)_22_	61.8	10	176–198	0.74	0.83	0.80
Mvi 1381	F: CCATCGGAGTTTCTCATCGT R: CCAGGTGCCCCTTACATT	(AC)_19_	61.8	7	185–197	0.53	0.76	0.73
Mvi 1843	F: AAATGGGAAGGTAAGGTAGAA R: CCTAAGGGACACAGACTTGC	(CA)_7_TA (AC)-	65.1	3	135–139	0.14	0.23	0.22
Mean	―	―	―	9	―	0.57	0.72	0.69

**Table 2 animals-11-00472-t002:** Genetic diversity indices and demographic characteristics for farmed and feral American mink in China.

Population.	Genetic Diversity Indices									Expansion Detection			Bottleneck Detection	
	Microsatellite						*Cyt-b*							
	*N*	*N_A_*	*AR*	*H_E_*	*Ho*	*F* _IS_	*h*	*Hd*	*π*	Tajima′s *D*	Fu′s *Fs*	Mismatch (*T*exp)	*P* _Wilcoxon test_	Mode Shift
Feral 1	10	4.92	4.08	0.62	0.55	0.19	4.00	0.71	0.01	−0.22	−5.99 *	No signal	0.924	Normal L-shaped
Feral 2	10	5.00	4.21	0.61	0.54	0.11	1.00	0.00	0.00	0.00	0.34	No signal	0.926	Shifted
Feral 3	6	4.50	4.50	0.66	0.63	0.06	4.00	0.80	0.01	0.27	−1.28	No signal	0.993	Normal L-shaped
Feral 4	6	4.08	4.08	0.62	0.50	0.21	6.00	1.00	0.02	−0.20	−0.43	No signal	0.995	Shifted
Farmed 5	16	6.33	4.64	0.66	0.59	0.10	3.00	0.24	0.00	−0.99	−23.63 *	N/A	0.788	Normal L-shaped
Farmed 6	14	6.58	4.94	0.71	0.60	0.17	8.00	0.82	0.01	−1.46	−8.52 *	N/A	0.849	Normal L-shaped

*N*, the number of analysed individuals for each marker; *N_A_*, number of alleles; *AR*, allelic richness; *H_E,_* expected heterozygosity; *H_O,_* observed heterozygosity; *F*_IS,_ inbreeding coefficients; *h*, number of haplotypes; *Hd*, haplotype diversity; *π*, nucleotide diversity. * means *p* < 0.05. N/A = least-squares procedure to fit the model of mismatch distribution and the observed distribution did not converge after 2000 steps.

## Data Availability

The mtDNA sequences and microsatellite data presented in this study are available on request from the corresponding author.

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
