# Peer review of "High Genetic Diversity of an Invasive Alien Species: Comparison between Fur-Farmed and Feral American Mink (Neovison vison) in China"

_animals, 2021, doi:10.3390/ani11020472_

Round 1
Reviewer 1 Report
I find that the revised paper is very precise and well written based on the materials, methods and results presented and thus recommend it for accept after a few additional considerations. My comments have been carefully considered and well responded to.
Only a few line specific comments remain for the annotated manuscript:
Line 85-87: Why 30 – 32 samples? Can you indicate or refer that is a sufficient number to describe genetic variability, or discuss it in the Discussion section?
I assume that colour type might affect some of the Genetic markers or Microsatellites, so could you please address colour type or state that it is not known/registered or that it will not have any impact.
It is not clear from the text if farmed mink were also sampled in 2011 – 2012, so maybe move this information to the beginning of the sentence, e.g.: ‘In 2011 to 2012, a total of 30 muscle tissue samples…..’
Line 178: In the revised sentence: ‘which correspond to that farmed and feral mink’, it seems that ‘that’ should be deleted or be ‘that of’?
Reviewer 2 Report
The manuscript describes population genetics analyses of farmed and feral American mink populations in China using mtDNA and STR markers. Although the geographic scale is limited, the results presented may be interesting for researchers in the field.
I suggest a grammar and spell check, as there are some grammatically incorrect, confusing or even incomplete sentences in the manuscript.
I also recommend to restructure the manuscript, and change the order of some paragraphs to follow the same order of Genetic diversity, Individual assignments and Population demography in all the Materials and Methods, Results and Discussion sections.
The authors generally refer to feral mink as one single population. Although the genetic structuring supports this grouping, the geographic distance between sampling sites makes the using of one single population questionable. Likewise, I doubt that samples originating from two different fur farms can be referred as a single population. It would sound better to change the confusing wording to simply feral/farmed mink or something else than population in this cases. Or the reasons of this grouping should be described in detail.
Line 16: Change “microsatellite sites” to “microsatellite loci” or “microsatellite markers”.
Lines 68-69: It seems like a verb is missing from the first part of the sentence.
Line 80-81: This sentence could be refined.
Line 90: Consider “hunted animals” instead of “hunted mink”. Generally, mink (farmed mink / feral mink) is used a lot repeatedly. I would suggest to change the wording where possible.
Line 91: Change “before DNA extraction” to “until DNA extraction”.
Line 98: Unnecessary and confusing wording, delete “samples of feral and farmed mink”.
Line 107: Instead of simply mtDNA be more precise, like "... fragment of the mitochondrial DNA containing the cytochrome b gene was amplified..."
Line 117: Use samples instead of mink.
Line 141: Denote the gene consequently, please.
Line 162: Please be more specific how samples were grouped for these calculations.
Lines 177-181: This part could be refined. For example: “Our clustering result supported the grouping of American mink samples from northeastern China into two clusters. One corresponding to samples of the farmed animals and the other to the feral samples. Some samples showed signs of admixture, as one farmed individual and five feral samples were assigned to their cluster with likelihoods less than 0.4.” Why is this cut-off value selected? There are more samples showing admixture with likelihoods somewhere between 0.3-0.7.
Line 202: Mixed haplotypes? I guess shared haplotypes were meant, but the sentence should be checked for clarity.
Lines 206 and 212: Use farmed mink consequently.
Line 215: Consider using “Genetic diversity”.
Table 1 and Table 2 should change places.
Lines 245-248: Samples of feral mink were grouped together and samples of farmed animals were also grouped. Why were samples of feral animals handled as a single population? Why were farmed mink from two different farms grouped together? Why weren’t the samples grouped according to sampling sites?
Line 256: Does populations refer to sampling sites or farmed vs. feral animals? The wording is confusing, see my general comments. Tajima’s is misspelled.
Lines 257-262: This part is confusing, it should be checked for clarity.
Line 268: This sentence is unnecessary, it should be deleted.
Lines 271-273: The word “samples” should be used instead of populations; but the sentence in this form makes no sense.
Lines 276-278: This sentence makes no sense. I suggest avoiding long and complicated sentences. It is easier to compose short sentences clearly.
Lines 283-285: This sentence could use some rewording. I guess I understand what the authors meant, but the grammar and the wording is awkward.
Lines 290-298: Did the studies mentioned use cytb sequences? The use of other genes or non-coding sequences may affect results.
Line 309: Mixed haplotypes?
Lines 316-320: Again, samples of feral mink were grouped together. If the mating of animals from different sampling sites is impossible, or at least very unlikely, grouping them into one single population may be misleading for these analyses and comparisons. The grouping of animals from two different farms together to detect inbreeding or estimate effective population size is also questionable. Samples should be grouped according to sampling sites, or the reasons of this grouping should be detailed.
Reviewer 3 Report
The Introduction is clear and useful, the laboratory and statistic methods appropriate and well-explained and the data analyses accurate. The text is well written and precise and the errors in English and editing should easily be caught by the authors in the revised form.
However, sampling, novelty of techniques and results’ output do not justify the publication of a full paper but rather a short communication.
Sampling seems to be an issue. For such a study, the number of the individuals examined, even combined with the use of two genetic markers, is in the edge of acceptable, preventing, in several occasions, the authors to conclude safely. I don’t think that it is possible to conduct a comprehensive and sound population genetic analysis using about 60 samples spread in 6 localities, υypothetically derived from different genetic backgrounds. Therefore, many of the conclusions lack serious statistical support.
In conclusion, the merit of the present work is limited in the genetic analysis for the first time of farmed and feral American mink populations introduced in China. However, small sample sizes not allowed any argumentation about the factors provoking genetic structuring. The authors should either improve sampling or give the present information in a short communication.
Round 2
Reviewer 2 Report
The authors implemented recommendations and the manuscript has been greatly improved as compared to the previous version.
Lines 76-78: This two statements somehow disrupt the line of thought. I would suggest to use something like “Although the number of samples is low, fifteen to twenty individuals per genetic cluster are sufficient to accurately estimate genetic diversity [19]. Additionally, there are different colour types of mink, but these did not affect the cytochrome b (Cyt-b) gene or the microsatellites used in this study [20].”
Lines 86-87: If the manufacturer’s protocol was followed, and overnight incubation is in the protocol described, this sentence is unnecessary.
Lines 134-158: I guess this part is deleted, including the equation.
Line 175: Change “For the entire population” to “For the whole sample set”.
Lines 178: Use the plural “feral populations”.
Line 179-180: Use “farmed animals” or “farmed mink” and the plural “feral populations”. Additionally, change “Both populations” to “All populations”.
Lines 186-187: Change the caption of Table 2 to something like ‘Genetic diversity indices and demographic characteristics…’.
Table 2: Change “clusters” to “population”.
Lines 197-198: Use the plural “feral populations” and “farmed mink”.
Lines 216-217: I would merge these two sentences: “The Bayesian phylogeny displayed five branches in the American mink from north-eastern China (Figure 3).” Additionally, I would suggest to refer to the branches of the phylogenetic tree not as genetic clusters.
Line 273: The word populations is repeated.
Line 277: Use simply “genetic diversity” without “level”.
Line 285: Feral in “feral: HO = 0.50-0.63, HE = 0.61-0.66” is unnecessary.
Author Response
Please see the attachment

This manuscript is a resubmission of an earlier submission. The following is a list of the peer review reports and author responses from that submission.
Round 1
Reviewer 1 Report
The paper presents an interesting study on the genetic resemplance between farmed and feral mink in China. It is novel and well written but some changes are needed, especially on the discussion, that should be restricted to the scientific results presented and their relevance and effects. Thus, the discussion of the impact of mink on the Chinese ecosystem must be deleted as it has no relevance relating to the results presented.
When the manuscript has been revise, the summary and abstract should be revised accordingly.
General comments:
The plural of mink in scientific literature is 'mink'. The plural form 'minks' may not be directly incorrect, but to me it sounds odd and unusual in a scientific paper. I suggest to write ‘mink’ all through the paper.
The American mink is from America and this is where it is wild. Other populations around the world are originating from farmed mink and are thus not 'wild' but 'feral' mink. Change 'Wild mink' with 'Feral mink' all through the paper!
Line specific comments:
Line 58: ‘breeding American minks reached 750,000 individuals.’ probably should be ‘breeding American mink reached 750,000 females’
Line 61-62: The claim that: ‘Recently, a fair amount of American minks 61 escape or are deliberately released into the wild in northeastern China.’ is an unsubstantiated statement that will not do in a scientific paper. It needs a reference or some other form of solid justification.
Line 64-66: If this is the documentation for the above claim, then replace line 61-62 with this, making the lack of genetic information on farmed and feral mink the conclusion of the paragraph. Even so, it is not clear why such genetic information is needed based on the feral population of mink, so it would be good if you could develop that argument further.
Line 69-71: Rather than just addressing questions I suggest that you formulate your expected answer to the questions, based on the reasoning in the introduction.
Line 71-72: I suggest to delete this part as it is not included in your research and you have no results to present on this issue.
Line 75-77: More information on sampling is needed! Why 30 - is that generally a sufficient number to describe genetic variability? How many samples were taken from each farm and from which colour types? Where were the farms situated, compared to the sampled feral mink? (It seems that some information is given in the discussion, but it should be here instead – and more precise)
Line 160-187, Figure 1, 2, 3 and 4: The legends in all four figures as well as text inside the figures should be: 'Farmed American mink' instead of ‘Farming’ and 'Feral American mink' instead of ‘Wild’.
Line 186: ‘the small red dot’ I don't see a small red dot! Are you referring to the many, apparently red, diamonds?
Line 187: ‘Red circle areas’ I only see one red/orang area?
Line 213: ‘but only for Fu's’ Delete ’for’
Line 214: ‘therefore, the rapid expansion hypothesis was rejected’ Do not discuss, accept or reject hypotheses under results! This is for the discussion section! Please delete, and bring it up in the discussion instead.
Line 215-216: ‘thus, the rapid expansion hypothesis was again rejected.’ Do not discuss, accept or reject hypotheses under results! This is for the discussion section! Please delete, and bring it up in the discussion instead.
Line 227-229: It is very confusing and makes no sense to write: ‘Moreover, we highlighted the impact of mink on the native ecosystem and propose several measures to effectively manage and control wild mink populations in China. In the following, we will discuss these proposals in more detail.’ You have done nothing of the sort, so delete the sentence. In case your results may indicate anything about the subject, you might include one small paragraph about this in the last part of the discussion, but it does not seen to me that you have.
Line 235: It sounds very odd and quite disturbing that you write: ‘and capturing from wild to farms is ongoing.’ Do you have any documentation for this? Please explain in the best way possible to what extent this is actually happening.
Line 237: ‘wild mink originating from farmed mink, which occurred in a short period’ What is the reasoning or background for the claim that ‘it occurred in a short period’? I suggest that instead you discuss what your result can say about this?
Line 238: ‘Two pedigrees of different origins of wild mink individuals’. Feral mink are individuals so rephrase to: ‘Two pedigrees of different origins of feral mink’.
Line 239-241: ‘How did you rethink the sources of feral mink in the region? What is your new thoughts, what do your results lead you to conclude after your rethinking? Please tell the reader what you think now! Discuss YOUR results and discuss to which extent previous results are in agreement with yours or not. Does 39, 40 and 41 still seem plausible or not, based on your results? I miss your conclusion on this paragraph.
Line 252-254: It is possible, but why is it likely, as you have no information on 'multiple admixtures'. So, is it only a posibility? How does this discussion concur with the discussion in section 4.1?
Line 261–280: Discussion sections are about discussing your results against those of others, and the relevance and impact of this. This section 4.3 has nothing to do with your results and must thus be deleted. In case your results actually do have some relevance on this subject, discuss that. As it is now, it is only guesswork without scientific value. If you have particular interest in this subject, investigate it and write another paper on the results – in this paper it is not relevant!
Line 273-274 + 278: The discussion on mink and otters seems misconstrued, as mink are ill equipped to compete with otters as they are smaller and cannot catch fish anyway as good as otters (Clode, D., & Macdonald, D. W. (1995). Evidence for food competition between mink (Mustela vison) and otter (Lutra lutra) on Scottish islands. Journal of Zoology, 237(3), 435-444). Furthermore, the number of mink will significantly decline in numbers if otters are introduced (Your own reference no 55). It appears unscientific and biased not to cite this paper in this context!!
Line 281-287: Like the previous paragraph, this section is also not relevant to your results - delete and discuss your results instead!
Line 287-290 is a good example of what to include!
Line 291-294: this section is also not relevant to your results - delete or discuss your results instead!
Line 303: Delete this last sentence as you cannot conclude something that you have not demonstrated in your results and have not reasoned out in the discussion, based on your results!
Reviewer 2 Report
This study uses microsatellite markers (12 SSRs) and mitochondria (cytb) to compare the farmed and wild The American mink. But I personally think that this study needs at least a few improvements. For example, the wild populations in China originally came from breeding escapes, try to compare with other studies (for example the same microsatellite marker, or at least cytb) is better. Further analysis of the number of effective population sizes or population dynamics is necessary. The wild sampling sites is not clear, it's best to match them with actual maps. Since there is only one wild population and one farmed population, it is not really good and useful in population genetics study. I personally suggest that at least more farmed and wild populations are needed to make sense.